# FungiTastic: A multi-modal dataset and benchmark for image categorization

**Lukas Picek** [🍄], **Klára Janoušková** [♛], **Milan Šulc** [🍄], and **Jiří Matas** [♛],

[🍄]University of West Bohemia & INRIA, [♛]CTU in Prague, and [🍄]Second Foundation
{lukaspicek,milansulc01}@gmail.com, {janoukl1,matas@fel.cvut.cz}

## Abstract

We introduce a new, highly challenging benchmark and a dataset – FungiTastic – based on data continuously collected over a twenty-year span. The dataset originates in fungal records labeled and curated by experts. It consists of about 350k multi-modal observations that include more than 650k photographs from 5k fine-grained categories and diverse accompanying information, e.g., acquisition metadata, satellite images, and body part segmentation. FungiTastic is the only benchmark that includes a test set with partially DNA-sequenced ground truth of unprecedented label reliability. The benchmark is designed to support (i) standard close-set classification, (ii) open-set classification, (iii) multi-modal classification, (iv) few-shot learning, (v) domain shift, and many more. We provide baseline methods tailored for almost all the use-cases. We provide a multitude of ready-to-use pre-trained models on HuggingFace and a framework for model training. A comprehensive documentation describing the dataset features and the baselines are available at GitHub and Kaggle.

## 1   Introduction

Biological problems provide a natural, challenging setting for benchmarking image classification methods. Consider the following aspects inherently present in biological data. The species distribution is typically seasonal and influenced by external factors such as recent precipitation levels. Species categorization is fine-grained, with high intra-class and inter-class variance. The distribution is often long-tailed; for rare species, only a very limited number of observations is available. New species are being discovered, raising the need for the "unknown" class option. Commonly, the set of classes has a hierarchical structure, and different misclassifications may have very different costs. Think of mistaking a poisonous mushroom for an edible one, which is potentially lethal, and an edible mushroom for a poisonous one, which at worse means coming back with an empty basket. Similarly, needlessly administering anti-venom after making a wrong decision about a harmless snake bite may be unpleasant, but its consequences are incomparable to not acting after a venomous bite.

The properties of biological data listed above enable testing of, e.g., both open-set and closed-set categorization methods, robustness to prior and appearance domain shift, performance with limited training data, and dealing with non-standard losses. In contrast, most common benchmarks operate under the independent and identically distributed (i.i.d.) assumption, which is made valid by shuffling data and randomly splitting it for training and evaluation. In real-world applications, i.i.d data are rare since training data are collected well before deployment and everything changes over time [37].

Submitted to the 38th Conference on Neural Information Processing Systems (NeurIPS 2024) Track on Datasets and Benchmarks. Do not distribute.

Data sources play an important role in benchmarking. In the age of LLMs and VLMs trained on possibly the entire content of the internet at a certain point in time, it is critical to have access to new, "unseen" data to guarantee that the tested methods are not evaluated on data they have indirectly "seen", without knowing. Conveniently, many domains in nature are of interest to experts and the general public, who both provide a continuous stream of new and annotated data. The general public's involvement introduces the problem of noisy training data; evaluating robustness to this phenomenon is also of practical importance.

In the paper, we introduce **FungiTastic**, a comprehensive multi-modal dataset of fungi observations which takes advantage of the favourable properties of natural data discussed above. The fungi observations include photographs, satellite images, meteorological observations, segmentation masks, and textual metadata. The metadata enrich the observations with attributes such as the timestamp, camera settings, GPS location, and information about the substrate, habitat, and biological taxonomy. By incorporating various modalities, the dataset support a robust benchmark for multi-modal classi-fication, enabling the development and evaluation of sophisticated machine learning models under realistic and dynamic conditions.

Classification of data originating in nature, including images of birds [3, 35], plants [13, 15], snakes [6, 24], and fungi [25, 34], has been used for benchmarking machine learning algorithms in several Fine-Grained Visual Categorization challenges; for a summary, see Table 1. Most of the commonly used datasets are small for current standards; the number of classes is also limited. The performance is often saturated, reaching total accuracy between 85-95 %; see the rightmost column of Tab. 1. Typically, the datasets are solely image-based and focused on traditional image classification; few of them offer basic attributes in metadata. Moreover, many popular datasets suffer from specific problems, e.g., regional, racial and gender biases [32], errors in labels [33, 4], and are saturated in accuracy.

Table 1: **Common image classification datasets** selected according to Google Scholar citations. We list suitability for closed-set classification (C), open-set classification (OS), few-shot (FS), segmenta-tion (S), out-of-distribution (OOD) and multi-modal (MM) evaluation and modalities, e.g., images (I), metadata (M), and masks (S), available for training. $\forall = \{$C, OS, FS, S, OOD, MM$\}$

| Dataset + citations (2022-24) | Classes | Training | Test | Modalities I | M | S | Tasks | SOTA[†] Accuracy |
|---|---|---|---|---|---|---|---|---|
| Oxford-IIIT Pets [23] 1,060 | 37 | 1,846 | 3,669 | ✓ | – | – | C | 97.1 [12] |
| FGVC Aircraft [21] 1,190 | 102 | 6,732 | 3,468 | ✓ | – | – | C | 95.4 [2] |
| Stanford Dogs [17] 680 | 120 | 12,000 | 8580 | ✓ | – | – | C | 97.3 [2] |
| Stanford Cars [19] 2,060 | 196 | 8,144 | 8,041 | ✓ | – | – | C | 97.1 [20] |
| CUB-200-2011 [35] 1,910 | 200 | 5,994 | 5,794 | ✓ | ✓ | ✓ | C | 93.1 [7] |
| NABirds [33] 283 | 555 | 48,562 | - | ✓ | – | – | C, FS, MM | 93.0 [10] |
| PlantNet300k [14] 30 | 1,081 | 243,916 | 31,112 | ✓ | – | – | C | 92.4 [14] |
| ImageNet-1k [9] 21,200 | 1,000 | 1,281,167 | 100,000 | ✓ | – | – | C, FS | 92.4 [11] |
| iNaturalist [34] 727 | 5,089 | 579,184 | 95,986 | ✓ | – | – | C, FS | 93.8 [30] |
| ImageNet-21k [27] 456 | 21,841 | 14,197,122 | - | ✓ | – | – | C, FS | 88.3 [30] |
| DF20 [25] 42 | 1,604 | 266,344 | 29,594 | ✓ | ✓ | – | C | 80.5 [25] |
| DF20–Mini [25] 42 | 182 | 32,753 | 3,640 | ✓ | ✓ | – | C | 75.9 [25] |
| FungiTastic — | 2,829 | 433,701 | 91,832 | ✓ | ✓ | ✓ | $\forall$ | 75.3 |
| FungiTastic–Mini — | 215 | 46,842 | 10,738 | ✓ | ✓ | ✓ | $\forall$ | 74.8 |

**The key contributions of the proposed FungiTastic benchmark are:**

- It includes diverse data types, such as photographs, satellite images, meteorological observations, segmentation masks, and textual metadata, providing a rich, multi-modal benchmark.

- Each observation is annotated with attributes like timestamp, camera metadata, location (longitude, latitude, elevation), substrate, habitat, and biological taxonomy, facilitating detailed studies and advanced classification tasks.

- It addresses real-world challenges such as domain shifts, open-set, and few-shot classification, providing a realistic benchmark for developing robust machine learning models.

- The dataset supports various evaluation protocols, including standard classification with novel-class detection, non-standard cost functions, time-sorted data for test-time adaptation methods, and few-shot classification.

- The test and validation data have not been published before and thus remain unseen by large language models (LLMs) and vision-language models (VLMs), maintaining the integrity and robustness of the evaluation process.

## 2 The FungiTastic Dataset

FungiTastic fungi is built on top of selected observations submitted to the Atlas of Danish Fungi before the end of 2023. Each observation includes at least one photograph and it is accompanied by additional metadata, see Figure1. In total, there are more than 650k images from 350k observations. The metadata include a multitude of attributes such as the timestamp, the camera settings, location (longitude, latitude, elevation), substrate, habitat, and taxonomy label. Not all observations have all of the attributes annotated, but the species attribute, which forms the basis for the primary classification task, has been annotated for all of the observations. Additionally, many images feature body-part segmentation masks and are supplemented by satellite images or meteorological data.

**Temporal division** reflecting the natural seasonality in fungi distribution is provided to ensure a standardized approach for training and model evaluation. The **FungiTastic–train** dataset consists of all observations up to the end of 2021[1], the **FungiTastic–val** and **FungiTastic–test** datasets encompass all observations from 2022 and 2023, respectively.

We define two types of classes, "unknown," with no examples in the training set; the remaining classes are tagged "known". The unknown classes are used in evaluations of open-set recognition. The open-set classification tasks are challenging as many of the unknown species look similar to the known ones. The closed-set validation and test sets include only classes present in the training set.

---

[1]the DF20 [25] training set with observations until the end of 2020 is a subset

user provided photographs (the knife left for scale)       satellite image

| | | | |
|---|---|---|---|
| Date: *2023-10-13* | Habitat: *Natural grassland* | Substrate: *Soil* |
| Location: *56.84, 9.01* | Taxon label: *Agaricus fissuratus* | Elevation: *28.5m* |

Figure 1: **An observation in FungiTastic** includes one or more images of a specimen (three leftmost columns) and possibly some of its parts, such as the microscopic image of its spores (second from the right). Metadata available for virtually all observations are listed at the bottom. Geospatial information is available for all observations (right), DNA sequencing for a subset.

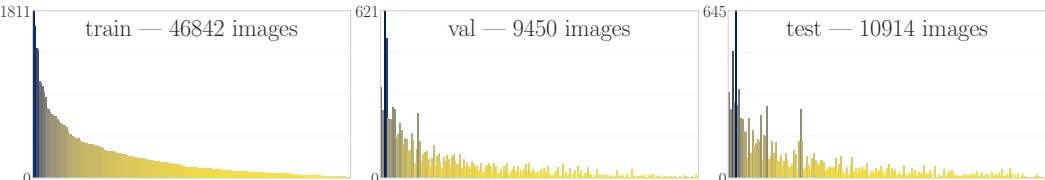

Figure 2: **Long-tailed distribution of classes (species) in the FungiTastic–M dataset** sorted by frequency on the training set and color-coded by the set frequency, showing class prior shift between the training, test, and validation sets. The classes are sorted by their frequency on the training set. The number of classes in these sets are 215, 193, and 196, respectively. Best viewed in Zoom.

**FungiTastic–M**, where M is for mini, is a compact and challenging subset of the FungiTastic dataset consisting of all observations belonging to 6 hand-picked genera primarily targeted for prototyping. These genera form fruit bodies of the toadstool type with a large number of species. The FungiTastic–M comprises 46,842 images (25,786 observations) of 215 species, greatly reducing the computational requirements for training. The training, validation and test splits are the same as for the full dataset. The long-tailed class (species) distribution can be seen in Figure 2.

**FungiTastic–FS** subset, FS for few-shot, is formed by species with less than 5 observations in the training set, which were removed from the main dataset. The subset contains 4,293 observations encompassing 7,819 images of a total of 2,427 species. As in the FungiTastic – closed set data, the split into validation and testing is done according to the year of acquisition.

Quantitative information about the FungiTastic is overviewed in Table 2.

Table 2: **FungiTastic dataset and benchmarks – statistical overview.** We provide the number of observations, images, and classes for each benchmark and the corresponding dataset. "Unknown classes" are those with no available data in training. DNA stands for DNA-sequenced data.

| Dataset | Subset | Observations | Images | Classes | Unknown classes | Metadata | Masks | Microscopic |
|---|---|---|---|---|---|---|---|---|
| FungiTastic – Closed Set | Train. | 246,884 | 433,701 | 2,829 | — | ✓ | – | ✓ |
| | Val. | 45,616 | 89,659 | 2,306 | — | ✓ | – | ✓ |
| | Test. | 48,379 | 91,832 | 2,336 | — | ✓ | – | ✓ |
| | DNA | 2,041 | 5,117 | 725 | — | ✓ | ✓ | |
| FungiTastic–M – Closed Set | Train. | 25,786 | 46,842 | 215 | — | ✓ | ✓ | ✓ |
| | Val. | 4,687 | 9,412 | 193 | — | ✓ | ✓ | ✓ |
| | Test. | 5,531 | 10,738 | 196 | — | ✓ | ✓ | ✓ |
| | DNA | 211 | 645 | 93 | — | ✓ | ✓ | ✓ |
| FungiTastic–FS – Closed Set | Train. | 4,293 | 7,819 | 2,427 | — | ✓ | – | ✓ |
| | Val. | 1,099 | 2,285 | 570 | — | ✓ | – | ✓ |
| | Test. | 998 | 1,909 | 566 | — | ✓ | – | ✓ |
| FungiTastic – Open Set | Train. | 246,884 | 433,701 | 2,829 | — | ✓ | – | ✓ |
| | Val. | 47,453 | 96,756 | 3,360 | 1,054 | ✓ | – | ✓ |
| | Test. | 50,085 | 97,551 | 3,349 | 1,013 | ✓ | – | ✓ |
| FungiTastic–M – Open Set | Train. | 25,786 | 46,842 | 215 | — | ✓ | – | ✓ |
| | Val. | 4,703 | 9,450 | 203 | 10 | ✓ | – | ✓ |
| | Test. | 5,587 | 10,914 | 230 | 34 | ✓ | – | ✓ |

## 2.1 Additional observation data

For approximately 99% of the image observations, visual data is accompanied by metadata, which includes information on environmental attributes, location, time, and taxonomy. This metadata is usually provided directly by citizen scientists and enables research on combining visual data with metadata. We provide around ten frequently completed attributes (see Table 3 for their description), with the most important ones listed and described below. Apart from the photographs and metadata provided by citizen scientists, we provide a wide variety of additional variables such as satellite images, meteorological data, segmentation masks, and textual metadata. In this section, we briefly describe the acquisition process for the most important one, and we provide.

Table 3: **Available metadata**. For all observations, we provide a comprehensive set of annotations. For species identification, the metadata allows to improve accuracy; see [10, 25].

| Metadata | Description |
|---|---|
| **Date of observation** | Date when the specimen was observed in a format yyyy-mm-dd. Besides, we provide three additional columns with pre-extracted *year*, *month*, and *day* values. |
| **EXIF** | Camera device attributes extracted from the image, e.g., metering mode, color space, device type, exposure time, and shutter speed. |
| **Habitat** | The environment where the specimen was observed. Selected from 32 values such as Mixed woodland, Deciduous woodland etc. |
| **Substrate** | The natural substance on which the specimen lives. A total of 32 values such as Bark, Soil, Stone, etc. |
| **Taxonomic labels** | For each observation, we provide full taxonomic labels that include all ranks from species level up to kingdom. All are available in separate columns. |
| **Location** | Location data are provided in various formats, all upscaled from decimal GPS coordinates. Besides the latitude and longitude, we also provide administrative divisions for regions, districts, and countries. |
| **Biogeographical zone** | One of the major biogeographical zones, e.g., Atlantic, Continental, Alpine, Mediterranean, and Boreal. |
| **Elevation** | Standardized elevation value, i.e., height above the sea level. |

**Meteorological Data**, i.e., climatic variables are vital assets for species identification and distribution modeling [1, 16]. In light of that, we provide 20 years of historical time-series values (2000 - 2020) of mean, minimum, and maximum temperature and total precipitation for all observations. We also provide an additional 19 annual average variables (temperature, seasonality, etc., averaged from 1981 to 2010). All the data was extracted from climatic rasters available at Chelsa.

**Remote sensing data** such as satellite images offer detailed and globally consistent environmental information at a fine resolution, making it a valuable resource for identification and other recognition tasks. To test the impact of such data and to facilitate easy use of geospatial data, we provide RGB satellite images in 128×128 pixel resolution (10m spatial resolution per pixel), centered on observation sights. The images were cropped out from rasters publicly available at Ecodatacube. As the raster's raw pixel values might include extreme values, we had to process the data further to be in a standardized and expected form. First, we clipped the values at 10,000. Next, the values were rescaled to a [0, 1] range and adjusted with a gamma correction factor of 2.5 (i.e., the values were raised to the power of 1/2.5). Last but not least, the values were rounded and rescaled to [0, 255].

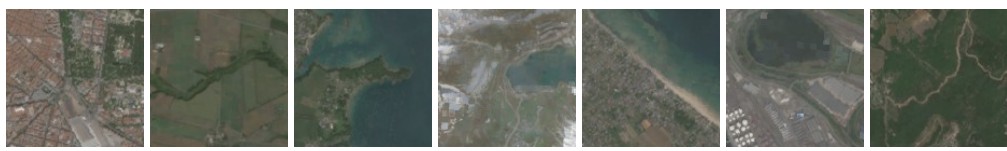

Figure 3: **Satellite images**. RGB images with a 128×128 resolution extracted from Sentinel2 data.

**Body part segmentation masks** of fungi fruiting body are essential for accurate identification and classification. These morphological features provide crucial taxonomic information distinguishing some visually similar species. Therefore, we provide human-verified instance segmentation masks for all photographs in the Funtastic mini dataset. We consider various semantic categories such as caps, gills, pores, rings, stems, etc. These annotations are expected to drive advancements in interpretable recognition methods [28], with masks also enabling instance segmentation for separate foreground and background modeling [5]. All segmentation mask annotations were semi-automatically generated in CVAT using the Segment Anything Model [18].

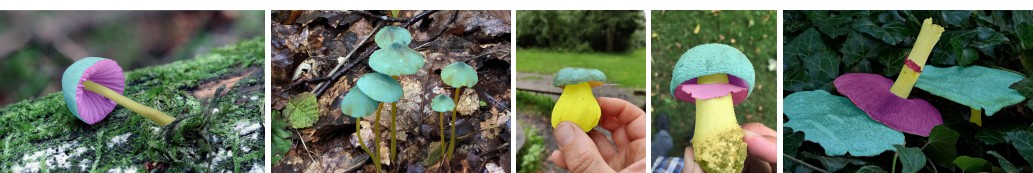

Figure 4: **Fruiting body part segmentation**. We consider cap, gills, stem, pores, and ring.

## 3 Challenges and evaluation

The diversity and unique features of the FungiTastic dataset allow for the evaluation of various fundamental computer vision and machine learning problems. We propose four distinct challenges, each with its own evaluation protocol. The remainder of this section is dedicated to a detailed description of each challenge and the associated evaluation metrics:

- Fine-grained closed-set classification with heavy long-tailed distribution – Subsection 3.1.
- Standard closed-set classification with out-of-distribution (OOD) detection – Subsection 3.1.
- Classification with non-standard cost functions – Subsection 3.3.
- Classification on a time-sorted dataset for benchmarking adaptation methods – Subsection 3.2.
- Few-shot classification of species with a small number of training observations – Subsection 3.4.

### 3.1 Closed and open set classification

In closed-sed classification, the set of classes in training and evaluation are the same while open set classification addresses scenarios where the input may belong to an unknown category that was not available during training. In FungiTastic, new species are being added to the database over time, including newly discovered species. The goal of closed-set classification is to develop a model that can classify inputs into known categories while open-set classification requires a model that can also identify inputs that do not belong to any of the known categories.

**Evaluation:** The main evaluation metric is $F$, the macro-averaged F1-score. For closed-set classification, the evaluation is standard, and for open-set, it is defined as

$$F = \frac{1}{C} \sum_{c=1}^{C} F_c, \quad F_c = \frac{2 P_c \cdot R_c}{P_c + R_c}, \tag{1}$$

where $P_c$ and $R_c$ are the recall and precision of class $c$ and $C$ is the total number of classes, including the unknown class $u$.

The F1-score of the unknown-class, $F^u$, and the F-score over the known classes, $F_k$, are also of particular interest, with $F_k$ defined as

$$F_K = \frac{1}{|K|} \sum_{c \in K} F_c, \tag{2}$$

where $K = \{1 \dots C\} \backslash \{u\}$ is the set of known classes. The $F_k$ also corresponds to the main evaluation metric for standard closed-set classification. Additional metrics reported are top-1 and top-3 accuracy, defined as

$$Acc@k = \frac{1}{N} \sum_{i=1}^{N} \mathbf{1}\left(y_i \in q_k(x_i)\right), \tag{3}$$

where $N$ is the total number of samples in the dataset, $x_i, y_i$ are the $i$-th sample and its label and $q_k(x)$ are the top $k$ predictions for sample $x$.

## 3.2 Temporal Image Classification

Each observation in the FungiTastic (FungiTastic) dataset is associated with a timestamp, enabling the study of how the distribution of different species evolves over time. The distribution of fungi species is seasonal and influenced by weather conditions, such as the amount of precipitation in previous days. Images from new locations may be included over time. This presents a unique real-world benchmark for domain adaptation methods, in particular online, continual and test-time adaptation. The challenge test dataset comprises images of fungi ordered chronologically. Consequently, a model processing an observation with a timestamp $t$ has access to all observations with timestamp $t'$ where $t' < t$.

**Evaluation:** The evaluation metrics are the same as those for the open-set recognition problem.

## 3.3 Classification beyond 0-1 loss function

Evaluation of classification networks is typically based on the 0-1 loss function, such as the mean accuracy, which applies to the metrics defined for the previous challenges as well. In practice, this often falls short of the desired metric since not all errors are equal. In this challenge, we define two practical scenarios: In the first scenario, confusing a poisonous species for an edible one (false positive edible mushroom) incurs a much higher cost than that of a false positive poisonous mushroom prediction. In the second scenario, the cost of not recognizing that an image belongs to a new species should be higher.

**Evaluation:** A metric of the following general form should be minimized

$$\mathcal{L} = \frac{1}{N} \sum_{i=1}^{N} W(y_i, q_1(x_i)), \tag{4}$$

where $N$ is the total number of samples, $(x_i, y_i)$ are the $i$-th sample and its label, $q_1(x)$ is the top prediction for sample $x$ and $W \in \mathbb{R}^{C \times C}$ is the cost matrix, $C$ being the total number of classes. For the poisonous/edible species scenario, we define the cost matrix as

$$W^{p/e}(y, q_1(x)) = \begin{cases} 0 & \text{if } d(y) = d(q_1(x)) \\ c_p & \text{if } d(y) = 1 \text{ and } d(q_1(x)) = 0, \\ c_e & \text{otherwise} \end{cases} \tag{5}$$

where $d(y), y \in C$ is a binary function that indicates dangerous (poisonous) species ($d(y) = 1$), $c_p = 100$ and $c_e = 1$. For the known/unknown species scenario, we define the cost matrix as

$$W^{k/u}(y, q_1(x)) = \begin{cases} 0 & \text{if } y = q_1(x) \\ c_u & \text{if } y = u \text{ and } q(x) \neq u, \\ c_k & \text{otherwise} \end{cases} \tag{6}$$

where $c_u = 10$ and $c_k = 1$.

## 3.4 Few-shot classification

Not only is the presented dataset highly imbalanced and the rarest species have as few as 1 observations, new species are also discovered and added over time. A few-shot segmentation approach based on, i.e., metric learning may be preferable both in terms of computational efficiency (retraining/finetuning the classifier to incorporate new species may be expensive) and accuracy.

For these reasons, we exclude the species with less than k observations from the main training set and provide a dedicated sub-dataset, the FungiTastic–FS.

**Evaluation:** Since the few-shot dataset does not have a severe class imbalance like the other FungiTastic subsets, this benchmark's main metric is top-1 accuracy. The F-1 score and top-k total accuracy are also reported. This challenge does not have any "unknown" category.

## 4 Baseline Experiments

In this section, we provide a variety of strong baselines based on state-of-the-art architectures and methods for three FungiTastic benchmarks. A set of pre-trained models was trained (inferred in the case of the few-shot classification) and evaluated on the relevant FungiTastic benchmarks. Bellow, we report results only for the closed-set and few-shot learning, but other baselines will be provided later in the supplementary materials, in the documentation, or on the dataset website.

### 4.1 Closed-set image classification

We train a variety of state-of-the-art CNN architectures to establish strong baselines for closed-set classification on the FungiTastic and FungiTastic–M. All selected architectures were optimized with Stochastic Gradient Descent, SeeSaw loss [36], momentum set to 0.9 and a mini-batch size of 64 for all architectures, and a learning rate of 0.01 (except ResNet and ResNeXt where we used LR=0.1), which was scheduled based on validation loss. While training, we used a Random Augment [8] data augmentation with a magnitude of 0.2.

Similarly to other fine-grained benchmarks, while the number of params, complexity of the model, and training time remain more or less the same as in convnets, the transformer-based architectures achieved considerably better performance on both FungiTastic and FungiTastic–M and two different input sizes (see Table 4.1). The best performing model, BEiT-Base/p16, achieved $F_1^m$ just around 40% which show severe difficulty of proposed benchmark.

Table 4: **Closed-set fine-grained classification FungiTastic and FungiTastic–M** A set of selected state-of-the-art CNN- (top section) and Transformer-based (bottom section) architectures. All reported metrics show the challenging nature of the dataset. The best result for each metric is **highlighted**.

| Architectures | FungiTastic–M – 224×224 | | | FungiTastic – 224×224 | | | FungiTastic–M – 384×384 | | | FungiTastic – 384×384 | | |
|---|---|---|---|---|---|---|---|---|---|---|---|---|
| | Top1 | Top3 | $F_1^m$ | Top1 | Top3 | $F_1^m$ | Top1 | Top3 | $F_1^m$ | Top1 | Top3 | $F_1^m$ |
| ResNet-50 | 61.7 | 79.3 | 35.2 | 62.4 | 77.3 | 32.8 | 66.3 | 82.9 | 39.8 | 66.9 | 80.9 | 36.3 |
| ResNeXt-50 | 62.3 | 79.6 | 36.0 | 63.6 | 78.3 | 33.8 | 67.0 | 84.0 | 39.9 | 68.1 | 81.9 | 37.5 |
| EfficientNet-B3 | 61.9 | 79.2 | 36.0 | 64.8 | 79.4 | 34.7 | 67.4 | 82.8 | 40.5 | 68.2 | 81.9 | 37.2 |
| EfficientNet-v2-B3 | 65.5 | 82.1 | 38.1 | 66.0 | 80.0 | 36.0 | 70.3 | 85.8 | 43.9 | 71.6 | 84.4 | 40.7 |
| ConvNeXt-Base | 66.9 | 84.0 | 41.0 | 67.1 | 81.3 | 36.4 | 70.2 | 85.7 | 43.9 | 71.2 | 84.2 | 40.0 |
| ViT-Base/p16 | 68.0 | 84.9 | 39.9 | 69.7 | 82.8 | 38.6 | 73.9 | 87.8 | 46.3 | 74.9 | 86.3 | 43.9 |
| Swin-Base/p4w12 | **69.2** | **85.0** | 42.2 | 69.3 | 82.5 | 38.2 | 72.9 | 87.0 | 47.1 | 74.3 | 86.4 | 43.1 |
| BEiT-Base/p16 | 69.1 | 84.6 | **42.3** | **70.2** | **83.2** | **39.8** | **74.8** | **88.3** | **48.5** | **75.3** | **86.7** | **44.5** |

Table 5: **Few shot classification on FungiTastic–Few-Shot**. (Left) – Pretrained deep descriptors with the nearest centroid and 1-NN nearest neighbor classification. All pre-trained models are based on the ViT-B architecture, CLIP, and BioCLIP with patch size 32 and DINOv2 with patch size 16. (Right)– Standard classification with cross-entropy-loss. Best result for each metric is **highlighted**.

| Model | Method | Top1 | $F_1^m$ | Top3 | Architecture | Input size | Top1 | $F_1^m$ | Top3 |
|-------|--------|------|---------|------|--------------|------------|------|---------|------|
| CLIP [26] | 1-NN | 6.1 | 2.8 | – | BEiT-B/p16 | 224×224 | 11.0 | 2.1 | 17.4 |
| | centroid | 7.2 | 2.2 | 13.0 | | 384×384 | 11.4 | 2.1 | 18.4 |
| DINOv2 [22] | 1-NN | 17.4 | 8.4 | – | ConvNeXt-B | 224×224 | 14.0 | 2.7 | 23.1 |
| | centroid | 17.9 | 5.9 | 27.8 | | 384×384 | 15.4 | 2.9 | 23.6 |
| BioCLIP [31] | 1-NN | 18.8 | **9.1** | – | ViT-B/p16 | 224×224 | 13.9 | 2.7 | 21.5 |
| | centroid | **21.8** | 6.8 | **32.6** | | 384×384 | 19.5 | 3.7 | 29.0 |

## 4.2 Few-shot image classification

Three baseline methods are implemented. The first baseline is standard classifier training with the Cross-Entropy (CE) loss. The other two baselines are nearest-neighbour classification and centroid prototype classification based on deep image embeddings extracted from large-scale pretrained vision models, namely CLIP [26], BioCLIP [31] and Dinov2 [22].

**Standard deep classifiers** are trained with the CE loss to output the class probabilities for each input sample. **Nearest neighbours classification (k-NN)** constructs a database of training image embeddings. At test time, $k$ nearest neighbours are retrieved and the classification decision is made based on the majority class of the nearest neighbours. **Nearest-centroid-prototype classification** constructs a prototype embedding for each class by aggregating the training data embeddings of the given class. The classification is performed based on the image embedding similarity to the class prototypes. These methods are inspired by prototype networks proposed in [29].

While DINOv2 [22] embeddings greatly outperform CLIP [26] embeddings, BioCLIP [31] (CLIP finetuned on biological data) outperforms them both, highlighting the dominance of domain-specific models. Further, the centroid-prototype classification always outperforms the nearest-neighbour methods in terms of accuracy, while nearest-neighbour wins over centroid-prototype in F-score. Finally, the best standard classification models trained on the in-domain few-shot dataset underperform both Dinov2 and BioCLIP embeddings in F-score, which shows the power of methods tailored to the few-shot setup. For results summary, refer to Table 5.

## 5 Conclusion

In this work, we introduced the FungiTastic, a comprehensive and multi-modal dataset and benchmark. The dataset includes a variety of data types, such as photographs, satellite images, meteorological observations, segmentation masks, and textual metadata. Biological data have many aspects interesting to the community such as long-tailed distribution or distribution shift over time. These aspects make the FungiTastic a rich and challenging benchmark for developing machine learning models.

The benchmark's challenging nature is demonstrated by classification-SOTA-based baselines. The best closed-set and few-shot classification models achieve an F-score of only 39.8 and 9.1, respectively, unlike many standard benchmarks, where state-of-the-art performance is approaching saturation.

**Limitations.** The data distribution is influenced by the data collection process, potentially introducing biases where certain species may be overrepresented due to their prevalence in frequently sampled areas or collector preferences. Nevertheless, we do not see how these biases could influence the image classification method evaluation. Additionally, not all meteorological data are available for every observation, which can affect of multi-modal classification approaches relying on such data.

**Future work** includes organizing ongoing challenges to monitor progress in image classification in various scenarios, regularly adding novel data and increasing the annotation coverage.

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
