# OpenReview forum: "FungiTastic: A multi-modal dataset and benchmark for image categorization"
_NeurIPS.cc/2024/Datasets_and_Benchmarks_Track — Submitted to NeurIPS 2024 Track Datasets and Benchmarks_

### Official Review · Reviewer_RwjS · 2024-07-11
**A multi-modal dataset and benchmark for image categorization**

**Rating:** 3
**Confidence:** 4
**Correctness:** The paper doesn't have any checklist,…
**Clarity:** The paper lacks clarity.

**Review:**

- The paper does not include a checklist!
- The link to Kaggle does not work.
- The github page looks empty, and I can't see code added to it.
- The contributions of the paper mentioned in lines 57-70:
  - The first and second, and possibly forth contributions are in fact one contribution (dataset).
  - The third contribution: 'It addresses real-world challenges such as domain shifts, open-set, and few-shot classification,
providing a realistic benchmark for developing robust machine learning models.', is not a paper contribution if the authors did not address them in their experiments!
  - I don't think the fifth contribution is a contribution either!

- Line 72-79 mentions that the "FungiTastic fungi is built on top of selected observations submitted to the Atlas of Danish Fungi before the end of 2023. ..." According to this section the observations and their corresponding attributes such as the timestamp, the camera settings, location (longitude, latitude, elevation), substrate, habitat, and taxonomy label are already available through the Atlas of Danish Fungi. Therefore, the authors must make it clear what novelty (new data or attributes), they are adding when publishing this dataset. It seems to me that the main contribution of the paper is to benchmark the dataset, and not the dataset itself. This needs to be clearly mentioned in the paper.
- For Introduction, the first 4 paragraphs are text descriptions without references.
- Table 1:
  - The paper requires comparing with more biological datasets of the recent years such as:
     - CWD30: https://arxiv.org/abs/2305.10084
     - BIOSCAN-1M: https://arxiv.org/abs/2406.12723
     - Insect-1M: https://arxiv.org/abs/2311.15206
     - Species-196: https://arxiv.org/abs/2309.14183

  - I don't think the 'citation column' is necessary and suitable for scientific papers.
  - Metadata column is confusing! What do you intend to show by metadata? if you want to mention the multimodal attributes of the dataset, then you should name them specifically, (for example taxonomic classification).
  - Why did you show both ImageNet-1k, and ImageNet-21k? How is it related to the dataset aspect and comparisons?
  - The STOA accuracy shows the accuracy of what task when for some of the dataset, there are multiple tasks?!
- Table2:
  - I don't understand what the authors intend to show by the metadata column?!
- The paper lacks dataset statistics, as it is unclear how much of the data has each attribute. It seems not all attributes are available for all data samples (stated in lines 76-77), and so this needs to be presented accurately.
- Figure 1:
  - The presentation requires improvement, the Microscopic image requires caption, and possibly all three images on the left.
  - If the first three images are for one fungi, I consider them as different samples of the same object so I don't think you need to show all samples.
- Are the satellite images captured from the collection site or the habitat of the Fungi?
  - I don't see how the satellite images are useful, especially when there are location (longitude, latitude, elevation), and habitat information of the organisms. The authors must make it clear how these images can be helpful.
  - It doesn't seem that the authors used the satellite images in the paper? Just as a dataset attributes?!
- Body part segmentation mask images.
  - Did the authors use masks in their experiments? It seems that they are presented as just dataset attributes?
  - I think by body part segmentation mask, the authors refer to the fungi physical traits, which their annotations are available for a small subset of the dataset.
- The paper presents experiments and results on close-set classification and few-shot classification tasks only.
  - I don't understand why the paper discusses concepts that are not addressed in the experiments including:
    - Open set classification
    - Temporal Image Classification
    - Classification beyond 0-1 loss function
 - What is the Section 3 about? Is it about the benchmark experiments of the submitted paper? It is too confusing!
- The overall presentation of the paper requires major works.

**Strengths:**

The Fungi dataset and its benchmarks offer valuable insights and applications across various domains, including biodiversity, ecosystem stability and resilience, precision agriculture, and food analysis.

**Additional Feedback:**

I have no additional feedback.

**Documentation:**

- The paper lacks checklist, and I am not sure if it should be subjected to desk reject.
- The link to Kaggle page does not work, and github page is empty by the time of this review.

**Ethics:**

The paper lacks checklist and therefore the license is not provided in the paper. This needs to be checked.

**Limitations:**

I addressed limitations under Review section.

**Opportunities For Improvement:**

- The introduction should become concise and cited.
- The related works requires revisions to include recent and related biological datasets, which I named a few of them.
- The presentation of the paper requires major works.
- The contributions of the paper are unclear and inaccurate the should be revised.
- The link to the Kaggle must be corrected.
- If the paper comes with a link to the github page, it need to be clean and complete, or at least work in progress.

**Relation To Prior Work:**

There is not enough discussion and comparison with related and recent biological datasets.

**Summary And Contributions:**

- The paper presents a diverse Fungi dataset with various data attributes.
- It discusses the potential applications and significance of the dataset.
- It designs and implements two sets of benchmark experiments using the dataset.

---

> ### Author Rebuttal · Authors · 2024-08-16
>
> > **C16: The paper presents experiments and results on close-set classification and few-shot classification tasks only. I don't understand why the paper discusses concepts that are not addressed in the experiments including: Open set classification, Temporal Image Classification, and Classification beyond 0-1 loss function. What is the Section 3 about? Is it about the benchmark experiments of the submitted paper? It is too confusing!**
>
> The main spirit of the dataset track is to provide data with interesting attributes that allow evaluation closer to the real-world setting. We are aware that we could add more experiments to the paper, and we will do it; however, in each case, we provide a protocol (data splits, evaluation metrics) for each case (classification 0-1, ….), which adds significant value to the paper (and that’s why the concepts are discussed).
>
> Here we add a citation from Joaquin Vanschoren and Serena Yeung [introduction to the NeurIPS Dataset and Benchmark Track.](https://neuripsconf.medium.com/announcing-the-neurips-2021-datasets-and-benchmarks-track-644e27c1e66c)
>
> _“There are no good models without good data [Sambasivan et al. 2021](https://research.google/pubs/pub49953/). The vast majority of the NeurIPS community focuses on algorithm design, but often can’t easily find good datasets to evaluate their algorithms in a way that is maximally useful for the community and/or practitioners. Hence, many researchers resort to data that are conveniently available, but not representative of real applications. For instance, many algorithms are only evaluated on toy problems, or data that is plagued with bias, which could lead to biased models or misleading results, and subsequent public criticism of the field [Paullada et al. 2020])(https://arxiv.org/abs/2012.05345).
> Researchers are often incentivized to benchmark their methods on a handful of popular datasets that have been well established in the field, with state-of-the-art results on these key benchmark datasets helping to secure a paper acceptance. Conversely, evaluations on lesser known real-world datasets, and other benchmarking efforts to connect models to real world impacts, are often harder to publish and are consequently devalued within the field.”_
>
> > **C17: The overall presentation of the paper requires major works.**
>
> We take your feedback on the presentation seriously and will work to improve the overall clarity and structure of the manuscript. However, we also want to highlight that the paper’s overall structure and readability were commended by other reviewers. Our revisions will aim to address your concerns while maintaining the positive aspects recognized by others.

---

> ### Author Rebuttal · Authors · 2024-08-16
>
> > **C7: Table 1: The paper requires comparing with more biological datasets of the recent years such as:
> CWD30, BIOSCAN-1M, Insect-1M, and Species-196.**
>
> We will update the table with recent biological datasets. We would like to point out that:
> CWD30 – is not a peer-reviewed paper; it is ArXiv only.
> BIOSCAN-5M –  is not a peer-reviewd paper, Is an ArXiv paper. Formatting suggests it is a concurrent submission to the NeurIPS 2024 Dataset track.
> Insect-1M –  is a CVPR24 paper published after the NeurIPS deadline.
>
> > **C8: I don't think the 'citation column' is necessary and suitable for scientific papers.**
>
> The citation info is a proxy for impact at the time of submission, which is of interest. It is easy to remove if the reviewers and area chairs prefer.
>
> > **C9: Metadata column is confusing! What do you intend to show by metadata? if you want to mention the multimodal attributes of the dataset, then you should name them specifically, (for example taxonomic classification).**
>
> We will address the lack of clarity of the 'metadata' column in Table 1 by removing unnecessary columns and clarifying the remaining ones. The intention was to provide comprehensive comparisons, and we will ensure this is presented more clearly.
>
> > **C10: Why did you show both ImageNet-1k, and ImageNet-21k? How is it related to the dataset aspect and comparisons?**
>
> Both are relevant as they are widely used in evaluating new methods. While ImageNet-1k includes “just” 1k categories and is saturated in performance, ImageNet-21k has 21k categories (i.e., is more fine-grained) and, therefore, is closer to the nature of our data. Besides, approximately 40% of the remaining mistakes on the ImageNet-1k can be classified as ‘major’ errors [Vasudevan et al.](https://arxiv.org/pdf/2205.04596) due to overfitting, which is still not the case for the ImageNet-21k.
> If the reviewers and area chairs prefer, the two lines in Table 1 with ImageNet-related info can be easily removed.
>
> > **C11: The SOTA accuracy shows the accuracy of what task when for some of the dataset, there are multiple tasks?!**
>
> That's a good point. The SOTA performance in Table 1 corresponds to the closed-set classification. As stated in lines 51-52, the point was to show the performance saturation of widely used datasets. The number of citations then represents how much they are used.
>
> > **C12: The paper lacks dataset statistics, as it is unclear how much of the data has each attribute. It seems not all attributes are available for all data samples (stated in lines 76-77), and so this needs to be presented accurately.**
>
> On line 100, we state that 99% of the observations are accompanied by all of the metadata – except for the masks, which are available only for the FungiTastic–M, which is stated in Table 2. Given the fact that it was not clear to the reviewer, we will include detailed statistics on the distribution of attributes across the dataset, specifying the availability of each attribute for the different subsets of data in the revised manuscript or documentation (based on available space).
>
> > **C13: Figure 1: The presentation requires improvement, the Microscopic image requires caption, and possibly all three images on the left.If the first three images are for one fungi, I consider them as different samples of the same object so I don't think you need to show all samples.**
>
> Can you please clarify why the microscopic image requires a caption same as the three images on the left? It seems obvious to other reviewers that it is a microscopic image. The consideration of different samples is incorrect. The goal is to classify it based on all the available data including multiple images of the same specimen, i.e., same mushroom.
> We will improve Figure 1 by providing clearer captions and ensuring that each image is labeled appropriately. We will also clarify whether the satellite images are directly tied to specific fungi collection sites or their general habitats, and discuss their relevance to the dataset's utility.
>
> > **C14: Are the satellite images captured from the collection site or the habitat of the Fungi?**
>
> As we state on lines 110-111:  “... we provide RGB satellite images in 128×128 pixel resolution (10m spatial resolution per pixel), centered on observation sights…”. the satellite images are related to the observation site (i.e., collection site).
>
> > **C14.1: I don't see how the satellite images are useful, especially when there are location (longitude, latitude, elevation), and habitat information of the organisms. The authors must make it clear how these images can be helpful.**
>
> Satellite images allow factors like proximity to a river or a road to be considered. This might be useful for detecting missing data, inaccuracies, or errors in the habitat attribute. Also, while the habitat information is categorical, the satellite image provides, in principle, more fine-grained information. Moreover, the habitat and substrate info is inserted manually, which is difficult for some citizen scientists.
>
> > **C14.2: It doesn't seem that the authors used the satellite images in the paper? Just as a dataset attributes?!**
>
> Indeed, we did not use it in the experiments section. However, as requested by you and other reviewers, we will include other experiments in the revised version. The results for the multi-modal approach and metadata exploiting are provided in the general response.
>
> > **C15: Body part segmentation mask images. Did the authors use masks in their experiments? It seems that they are presented as just dataset attributes?**
>
> We did not provide experiments with available masks as the annotation process finished close to the submission deadline, and we had to correct some small issues related to manual labeling. We are currently working on a baseline that uses the masks, and we will add it to the revised version or documentation.

---

> > ### Comment · Reviewer_RwjS · 2024-08-17
> > **Second Rebuttal**
> >
> > Thanks for the second rebuttal by authors. In the following, I am going to address a few points but I will consider the authors responses to all comments:
> >
> > - I meant BIOSCAN-1M, but apparently I put a link of BIOSCAN-5M.
> >    - BIOSCAN-1M: https://neurips.cc/virtual/2023/poster/73549  and  ahttps://arxiv.org/abs/2307.10455
> > - I don't think here we are looking for how much a work has been cited, since papers are published at different times, for various communities, and including different contributions their usage is not factor to consider.
> > - Captions: I think images and tables should be self explanatory. Thank you for pointing out that this is obvious to the remaining reviewers, however, if you want your work be usable for a broad community you may include information in the images.

---

> ### Author Rebuttal · Authors · 2024-08-16
>
> We appreciate your time and effort invested in reviewing the manuscript and the valuable feedback provided; thank you. Please find our response related to your comments bellow.
>
> >**C1: The paper does not include a checklist!**
>
> We are sincerely sorry for accidentally not compiling the checklist into the submission. The checklist is in the attached “rebuttal file.”
>
> >**C2: The link to Kaggle does not work.**
>
> We apologize for the confusion regarding the Kaggle link. The [Kaggle link](https://www.kaggle.com/datasets/c2a2f11e32d1dbe110965def355907965a28b05a83244285ce7b515d2f785b0f) is in the supplementary material, we did not link from the abstract and the datasheet, a mistake we’ll correct. The link has been tested across multiple devices, systems, and browsers
>
> >**C3: The github page looks empty, and I can't see code added to it.**
>
> Thank you for pointing out the issue with the GitHub link which arose when we moved the repository, which inadvertently broke the link provided in the paper. We realized this mistake a few weeks ago and added a redirection to the correct repository on July 28, which can be verified in the commit history. The correct link was included in the supplementary material and Datasheet, but we acknowledge that it was not updated in the main text of the paper. We will ensure that the revised version of the paper contains the updated and [correct GitHub link](https://github.com/bohemianvra/FungiTastic/) to prevent any further confusion.
>
> >**C4: The contributions of the paper mentioned in lines 57-70:**
>
> We will refine this section to more clearly distinguish between the novel aspects of our dataset construction and the benchmarking framework.
>
> >**C4.1: The first and second, and possibly forth contributions are in fact one contribution (dataset).**
>
> (i) We will merge the first two contributions, which indeed refer to observational data. The fourth contribution is different: we introduce evaluation protocols assessing properties of ML models that are important in applications but often ignored in research-oriented benchmarks
>
> >**C4.2: The third contribution: 'It addresses real-world challenges such as domain shifts, open-set, and few-shot classification, providing a realistic benchmark for developing robust machine learning models.', is not a paper contribution if the authors did not address them in their experiments!**
>
> (ii) The third contribution is in defining protocols that will help researchers properly assess robustness to domain shifts, deal with few-shot classification, etc. It is beyond the scope of the paper to address all these challenges and provide experiments and baselines.
>
> >**C4.3: I don't think the fifth contribution is a contribution either!**
>
> (iii) The fifth contribution – data unseen by LLMs – is important (since it guarantees that test data have not indirectly leaked into training) and not easy to achieve.
> We note that other reviewers found the contributions to be significant and the paper logically-structured, particularly noting the innovation in provided multimodal data and evaluation protocols.
>
>  > **C5: Line 72-79 mentions that the "FungiTastic fungi is built on top of selected observations submitted to the Atlas of Danish Fungi before the end of 2023. ..." According to this section the observations and their corresponding attributes such as the timestamp, the camera settings, location (longitude, latitude, elevation), substrate, habitat, and taxonomy label are already available through the Atlas of Danish Fungi. Therefore, the authors must make it clear what novelty (new data or attributes), they are adding when publishing this dataset. It seems to me that the main contribution of the paper is to benchmark the dataset, and not the dataset itself. This needs to be clearly mentioned in the paper.**
>
> While the FungiTastic dataset is based on existing observations from the Atlas, our work with data goes well beyond compiling this data. FungiTastic dataset novelties lie in an extensive enhancement and integration of multi-modal data sources and the creation of benchmark tasks tailored specifically for machine learning applications. To be more specific, we have:
> - Fixed errors in the provided Habitat and Substrate labels and updated them to a new hierarchical structure.
> - Added elevation which is not available in Atlas of Danish Fungi!
> - Added a species toxicity level.
> - Added higher location attributes such us Region, District and Country.
> - Added biogeographical region, e.g., Atlantic, Continental, etc.
> - Added Landcover classification.
> - Added Remote Sensing Data for each observation.
> - Added 20 years of Meteorological Data.
> - Manually labeled ~500k images to get a “micros.” tag.
> - Manually labeled ~60k images with body part image masks.
> - We have developed multiple splits based on natural evolution captured by the data. that form interesting ML problems.
>
> **Note 1:** We would like to stress that we have been working closely with the Atlas of Danish Fungi since 2018 [Sulc et al.](https://openaccess.thecvf.com/content_WACV_2020/papers/Sulc_Fungi_Recognition_A_Practical_Use_Case_WACV_2020_paper.pdf), [Picek et al.](https://www.mdpi.com/1424-8220/22/2/633), [Picek et al.](https://openaccess.thecvf.com/content/WACV2022/papers/Picek_Danish_Fungi_2020_-_Not_Just_Another_Image_Recognition_Dataset_WACV_2022_paper.pdf), and co-authors of this paper worked or works on the platform itself.
>
> **Note 2:** Considering a “compilation” of previously available data as not worthy of this track is not correct. The SCOPE of this track presented on the [web](https://nips.cc/Conferences/2024/CallForDatasetsBenchmarks) includes “New datasets, or carefully and thoughtfully designed (collections of) datasets based on previously available data.”
>
> >**C6: For Introduction, the first 4 paragraphs are text descriptions without references.**
>
> We will revise the introduction to include more citations, early in the text, that support the background and motivation for our work.

---

> > ### Comment · Reviewer_RwjS · 2024-08-17
> > **Third Rebuttal**
> >
> > Thanks for the responses.
> >
> >
> > - My point is not that **Considering a compilation of previously available data as not worthy of this track**.
> >
> >    - The submitted paper lacks detailed information about data collection including checklist, github page, and Kaggle link. We can not speculate the points that authors mentioned in their rebuttal despite lacking clear and accurate documentations, especially checklist, which is a mandatory requirement for submission by NeurIPS.
> >
> > - **For the second time the authors referenced other reviewers**
> >    - I am not sure what the authors intend to show by emphasizing on the evaluations of other reviewers!
> >    - The reviewers evaluate papers **independently** based on their own judgments, otherwise each paper would have not been given to multiple reviewers.
> >
> > - Regarding the following points, the new contents are as follows:
> >     - Geo info:
> >         - Fixed errors in the provided Habitat and Substrate labels and updated them to a new hierarchical structure.
> >              - *Are you referencing data cleaning and curation?*
> >         - Elevation which is not available in Atlas of Danish Fungi!
> >         - Location attributes: Region, District, Country, Atlantic, and Continent,
> >         - Added Remote Sensing Data for each observation.
> >
> >     - What do you mean by **adding Landcover classification**? *are these labels added to the samples?*
> >     - Added a species toxicity level. *Are these labels added to the samples?*
> >
> >
> >     - Added 20 years of Meteorological Data. *Did you describe what this inclusion is about in the submitted draft?*
> >
> >     - Exp info:
> >        - Manually labeled ~500k images to get a “micros.” tag.
> >        - Manually labeled ~60k images with body part image masks.
> >        - We have developed multiple splits based on natural evolution captured by the data. that form interesting ML problems.

---

> ### Comment · Reviewer_RwjS · 2024-08-17
> **First Rebuttal**
>
> - The track is dataset and benchmark, thus proposing new datasets and benchmarks is what is expected for this track. Thanks for the quotation from Joaquin Vanschoren and Serena Yeung. I assume for such reasons NeurIPS opened the dataset and benchmark track so that is not a matter of discussion and not my point.
>
> - As I mentioned in my comments you can not presents tasks and potential experiments in your paper as your contributions unless you have conducted such experiments and reported the results. Knowing the potential experiments with your dataset including Open set classification, Temporal Image Classification, and Classification beyond 0-1 loss function, are not your contributions (they are known to community). You may mention these potential paths for new tasks and experiments in conclusion and future works but not as your contributions. I am sure that the authors are aware that experiment-related contributions are counted for the actual designed and implemented experiments, and not just hypothesizing and discussing the potentials.

---

> > ### Author Response · Authors · 2024-08-18
> >
> > > **Knowing the potential experiments with your dataset including Open set classification, Temporal Image Classification, and Classification beyond 0-1 loss function, are not your contributions (they are known to community).**
> >
> > We are not claiming that we invented these classification problems. However, we are not aware of a non-synthetic dataset that has a defined protocol for these practically important tasks, including all the datasets mentioned in the paper and in the reviews. If you are aware of any such dataset and protocol, in the context of a challenging real-world fine-grained problem, please let us know, and we’ll refer to it.
> > If a practitioner wished to experimentally evaluate a novel method with temporal classification, classification beyond 0-1, or open set, they would have to do all the work – define the dataset, splits, and the protocol. FungiTastic is very helpful to the community in addressing this need. Moreover, it attracts attention to these neglected classification problems.
> >
> > > **As I mentioned in my comments you can not presents tasks and potential experiments in your paper as your contributions unless you have conducted such experiments and reported the results.**
> >
> > We do not agree with this view.  As explained above, the data and protocol are valuable per se. Providing a baseline experiment is easy, and it could be done by any user – the first results published on Kaggle will become a baseline.
> > We accept that we could have inserted the baselines in the supplementary.  We provide results of additional baselines in the General Response.

---

> > > ### Comment · Reviewer_RwjS · 2024-08-19
> > >
> > > Please refer to the information I provided below.
> > >
> > > **Open Set Classification**
> > >
> > > Open set classification deals with scenarios where the model encounters classes during testing that *were not present during training*. Unlike traditional classification, where the model is trained on a fixed set of classes and expects test data to belong to one of these classes, open set classification needs the model to recognize when a new, **unseen class** is encountered and respond accordingly (usually by rejecting the input or flagging it as unknown).
> > >
> > > - **Qualities a Dataset Should Have:**
> > >     - **Diverse and Incremental Classes:** The dataset should have a well-defined set of *known classes for training* and additional *classes for testing that were not seen during training*.
> > >     - **High Variability:** There should be a wide range of features within each class to ensure that the model can distinguish between known and unknown classes effectively.
> > >     - **Labeled and Unlabeled Data:** Labeled data for known classes is essential, along with a portion of unlabeled or unknown class data to simulate open set conditions.
> > >
> > > - **Example Datasets:**
> > >     - CIFAR-10/100: CIFAR-10 could be used as the known set, with CIFAR-100 classes serving as unknowns.
> > >     - ImageNet: Some classes can be held out during training and used as unknowns during testing.
> > >     - MNIST/SVHN: MNIST digits can be the known classes, while other digits or letters can be treated as unknowns.
> > >     - Biological datasets like BIOSCAN-1M, iNatrualist, Insect-1M, BenthicNet: These have hierarchical taxonomic classification labels at large scale, where at each level numerous classes (please look at the corresponding articles and their dataset statistics) can be divided into known and unknown classes. They also have numerous samples that are unlabelled.
> > >
> > > **Temporal Image Classification**
> > >
> > > Temporal image classification involves classifying images that have a *temporal or sequential aspect*, such as frames in a video or images taken over time. The goal is to consider not just the individual frames but also the temporal context in which they appear to improve classification accuracy.
> > >
> > > - **Qualities a Dataset Should Have:**
> > >
> > >    - **Temporal Sequence Data:** The dataset should consist of **sequences of images or frames**, where the temporal relationship between consecutive images is important.
> > >    - **Labeling of Sequences:** Each sequence or set of temporally-related images should have corresponding labels that may change over time.
> > >    - **High Frame Rate:** For video-based tasks, a high frame rate ensures that temporal changes can be effectively captured.
> > >
> > > - **Example Datasets:**
> > >    - UCF101: A dataset of action recognition from videos, useful for temporal image classification.
> > >    - Kinetics-700: A large-scale dataset for human action recognition in videos.
> > >    - KITTI: Contains sequences for autonomous driving scenarios, where temporal aspects are critical.
> > >
> > > **Classification Beyond 0-1 Loss Function**
> > >
> > > Classification beyond 0-1 loss refers to scenarios where the *standard accuracy metric (i.e., treating all misclassifications equally) is replaced by more nuanced loss functions*. These can include *weighted loss functions* where certain types of errors are penalized more heavily or losses that consider the hierarchical structure of classes. For example **Focal Loss** (used in experiments of BIOSCAN-1M) can be considered a type of **weighted loss function** that fits within the broader category of classification beyond the 0-1 loss function.
> > >
> > > - **Qualities a Dataset Should Have:**
> > >      - **Class Imbalance:** A dataset with imbalanced classes can highlight the need for more complex loss functions that penalize misclassifications differently.
> > >     - **Hierarchical Class Labels:** If the dataset has a hierarchy of classes (e.g., species, genus, family), this supports experiments with losses that respect class similarity.
> > >
> > >
> > > - **Example Datasets:**
> > >     - ImageNet (Hierarchical): The hierarchical nature of classes in ImageNet can be used for experiments with structured loss functions.
> > >     - CIFAR-100: With a broader set of classes, this dataset can be used for weighted loss functions that prioritize certain classes.
> > >     - Medical Imaging Datasets (e.g., CheXpert): These datasets often have class imbalances and can be used to experiment with loss functions that penalize false negatives more than false positives.
> > >     - Biological datasets like BIOSCAN-1M, iNatrualist, Insect-1M, BenthicNet: These have hierarchical taxonomic classification labels at large scale, with high degree of class imbalance ratio.

---

> > ### Author Response · Authors · 2024-08-19
> >
> > Dear Reviewer, we will be brief in our response.
> >
> > Notice how many datasets had to be listed, and how many times “can be” (12x) and “could be” (1x) had to be mentioned – i.e., it is not possible with the current state of the art – to achieve what FungiTastic has “all-in-one.”, thus providing the community with added value.
> >
> > By temporal image classification, we mean that all training data are older than all test data, just like in a deployed ML system in the real-world (which implies that the training and test data are not drawn from the same distribution).

---

> > > ### Comment · Reviewer_RwjS · 2024-08-26
> > >
> > > Thanks for your response and clarification.
> > >
> > >
> > >
> > > The term **temporal classification** is correct for describing the general process of classifying data over time, especially when dealing with time-series data or sequences. However, if you want to emphasize the difference between the training and test data distributions due to the time gap, **domain shift** is a more precise term. Domain shift refers to the situation where the statistical properties of the training data and the test data differ, which is often the case in temporal settings. So, if your focus is on the time aspect of the data, "temporal classification" is appropriate. But if you're emphasizing the challenge posed by the differing distributions due to time, "domain shift" would be the correct term to use.
> > >
> > >
> > > Based on my comments regarding the lack of thorough comparisons with related biological datasets, the absence of experiments and tasks that align with the dataset's potential and the paper's contributions (Open set classification, Temporal Image Classification, and Classification beyond 0-1 loss function), and the response to one of my comments about the dataset itself, I would maintain my original score.

---

### Official Review · Reviewer_3VQA · 2024-07-25
**Well thought-out paper and dataset**

**Rating:** 8
**Confidence:** 4
**Clarity:** The paper is clear and well written.

**Review:**

Overall, a good paper that I would be happy to accept. I do not see any major issues and do appreciate the interest in curating a dataset that provides some realistic value in terms of both biological studies and ML.

Pros:
- Well written.
- Well thought-out with clear explanations.
- Great motivation for Challenges (sect 3). Each of these relates to an interesting and pertinent aspect of the dataset and its realistic use. I especially appreciated the inclusion of "Classification beyond 0-1 loss function", as this greatly improves the realistic utility of the data.
- Processing of the data is adequately explained.
- Dataset is available on kaggle.

Cons:
- It would have been nice to see more categories of baseline experiments presented at time of review.
- It would have been nice to see experiments that make use of multi-modal nature of this data. I assume the experiments were run only on the fungi images, rather than involving other aspects such as spore or satellite images, DNA, etc?
- It seems code is not available.

**Strengths:**

Similar to 'pros' listed in main review. I think the main strengths of this paper are the thorough compilation of multi-model data and the explicit and well-motivated challenges proposed as benchmark tasks. This dataset, although biological, is not limited in its utility to biologists, rather, it presents relevant challenges to the broader ML community.

**Additional Feedback:**

None.

**Correctness:**

Claims seem to be correct to the best of my knowledge and understanding. The dataset seems to be constructed in a sound way. Benchmark evaluation methods are well thought out.

**Documentation:**

There is sufficient detail given on data collection and organization, availability and maintenance, and ethical and responsible use. See supplemental for details. Code for benchmark experiments would be appreciated.

**Ethics:**

No concerns.

**Limitations:**

Limitations are adequately discussed.

**Opportunities For Improvement:**

Similar to 'cons' in main review.

- It would have been nice to see more categories of baseline experiments presented at time of review. Please do this.
- It would have been nice to see experiments that make use of multi-modal nature of this data. I assume the experiments were run only on the fungi images, rather than involving other aspects such as spore or satellite images, DNA, etc?
- It seems code is not available. Please provide basic code for dataloading and for challenge-specific metrics.

**Relation To Prior Work:**

A comparison to previous biological image classification datasets is given.

**Summary And Contributions:**

The paper presents a large dataset of fungi (primarily from Denmark) and benchmarks for compelling challenges and for which the performance is not nearly at saturation (as is too often the case for benchmarks, making them uninteresting).

The dataset (constructed from samples of the Atlas of Danish Fungi submitted before the end of 2023) is multi-model, comprising camera photos of fungi, satellite images of the source location of these samples, meteorological observations, segmentation masks, and textual metadata. The metadata includes timestamp, camera settings, GPS location, and information about the substrate, habitat, and biological taxonomy. In some cases, DNA is included. The authors have partitioned the dataset based on the proposed benchmark tasks, supporting open-set classification, closed-set classification, few-shot classification/learning, and a miniature (but still highly challenging) version of the dataset. The proposed challenges (sect 3) delve into four compelling benchmarking tasks. These include close-set and open-set classification (dataset is always growing, therefore new categories would be unseen, similar to open-set classification), temporal classification (learning from past samples only, thereby mimicking the real-world evolution of the dataset), classification beyond 0-1 loss function (taking into consideration the practical importance and costs of correct/incorrect classification, such as in the case of determining whether something is poisonous), few shot classification (data is limited for rare species, so this is a relevant test). Experiments are performed for closed-set and few-shot image classification, with conclusions indicating that domain- and task-specific models tend to perform best. Limitations and a note on future work are given.

---

> ### Author Rebuttal · Authors · 2024-08-16
>
> Thank you for your time and valuable feedback. Please see our responses to your comments below.
>
> > **C1: It would have been nice to see more categories of baseline experiments presented at time of review. It would have been nice to see experiments that make use of multi-modal nature of this data. I assume the experiments were run only on the fungi images, rather than involving other aspects such as spore or satellite images, DNA, etc?**
>
> Indeed, we ran the experiment just with a single modality, i.e., provided photographs, including microscopic images. In the revised version of the paper and/or supplementary materials, we will include at least experiments related to metadata utilization and multi-modal learning; some tables are provided in the General Response. We are already working on the remaining baselines for most of the proposed benchmarks.
>
> > **C2: It seems code is not available. Please provide basic code for dataloading and for challenge-specific metrics.**
>
> We apologize for an issue related to the [GitHub link](https://github.com/bohemianvra/FungiTastic/). At the moment dataLoaders, custom metrics, and a preliminary version of the code are available. The final version will be provided after the review process; preliminary version of the code is available in feature branches which will be soon merged into dev/main.

---

> > ### Comment · Reviewer_3VQA · 2024-08-30
> >
> > Thank you for your response. This is all great! I am pleased to see that you will be adding more experiments. Looking forward to seeing this.

---

### Official Review · Reviewer_sU1H · 2024-07-29
**The paper introduces the FungiTastic dataset, which exhibits a high degree of innovation and challenge.**

**Rating:** 8
**Confidence:** 4
**Correctness:** Correct.
**Clarity:** Yes.

**Review:**

The article offers a variety of baseline methods, with pre-trained models available on HuggingFace, as well as a framework for model training. These baseline methods cover a range of use cases, including standard closed-set classification, open-set classification, multi-modal classification, few-shot learning, and domain shift, providing a robust reference for subsequent research. The article proposes four distinct challenges with corresponding evaluation protocols, including fine-grained closed-set classification, classification with non-standard cost functions, and classification on a time-sorted dataset. The design of these challenges and evaluation metrics is conducive to a comprehensive test and advancement of machine learning model performance. The article is written, logically structured, and provides readers with abundant information and in-depth analysis. It is anticipated that the dataset will inspire more innovative research work and promote technological progress in related fields.

**Strengths:**

The article offers a variety of baseline methods, with pre-trained models available on HuggingFace, as well as a framework for model training. These baseline methods cover a range of use cases, including standard closed-set classification, open-set classification, multi-modal classification, few-shot learning, and domain shift, providing a robust reference for subsequent research. The article proposes four distinct challenges with corresponding evaluation protocols, including fine-grained closed-set classification, classification with non-standard cost functions, and classification on a time-sorted dataset. The design of these challenges and evaluation metrics is conducive to a comprehensive test and advancement of machine learning model performance.

**Additional Feedback:**

N/A

**Documentation:**

N/A

**Ethics:**

No.

**Limitations:**

From my perspective, the authors have made a huge contribution by constructing such a dataset, which I think is of great value in advancing related research. The authors have addressed the limitations in the paper, so please continue to improve it in your subsequent work.

**Opportunities For Improvement:**

The authors discuss the limitations of the dataset at the end of the article, such as biases that may be introduced by the data collection process and incomplete meteorological data for some observations. At the same time, the authors propose future directions for work, including organizing ongoing challenges, regularly adding new data, and increasing annotation coverage. It is recommended that the authors refine these deficiencies and limitations in future work.

**Relation To Prior Work:**

Yes.

**Summary And Contributions:**

The paper introduces the FungiTastic dataset, which exhibits a high degree of innovation and challenge. The dataset comprises approximately 350,000 multi-modal observations, encompassing over 6,500 photographs across 5,000 fine-grained categories, as well as a wealth of ancillary information, such as acquisition metadata, satellite imagery, and body part segmentation. The benchmark includes a test set partially grounded in DNA sequencing, offering an unprecedented level of label reliability, which is of significant importance for advancing image classification technology. The paper provides a detailed description of the various data types included in the dataset, such as photographs, satellite imagery, meteorological observations, and textual metadata. This diversity provides a rich resource for research in multi-modal classification, domain adaptation, and few-shot learning. The construction of the FungiTastic dataset takes into account the inherent characteristics of biological data, such as seasonal distribution, fine-grained categorization, and long-tailed distribution, which endows the dataset with high practical value.

---

> ### Author Rebuttal · Authors · 2024-08-16
>
> Thank you for your positive and encouraging review. We are pleased that you found the FungiTastic dataset innovative and valuable for advancing research in multimodal classification, domain adaptation, and few-shot learning. We appreciate your recognition of the dataset's practical value, particularly its grounding in biological characteristics and its potential for diverse applications.
>
> We are committed to continually improving the dataset/benchmarks and providing new baselines to the community.

---

### Official Review · Reviewer_WN3u · 2024-07-30
**Review of paper #709**

**Rating:** 5
**Confidence:** 3

**Review:**

This paper introduces a large-scale multimodal dataset for fine-grained image categorization, specifically focused on fungi. However, it is crucial to verify the usefulness of the metadata claimed as a contribution in this paper. Additionally, there is a complete lack of analysis regarding the methodological approaches and a discussion of existing methods. To ensure the future utility of this dataset and the validation of methods across various tasks, performance analysis on multimodal data should be included.

**Strengths:**

- This paper is simple and easy to follow.
- It collects a lot of images and a lot of kinds of metadata.
- This paper introduces several tasks for real-world fungi recognition, not limited to a closed-set recognition task.

**Additional Feedback:**

- Line 141: "In closed-sed" should be "In closed-set."
- Line 153: F_k -> F_K

**Clarity:**

It is simple to follow, and they provide information about datasets through proper table organization.

**Correctness:**

Fungi classification would be useful for real-world applications. In addition to the mushroom image, there is supplementary information that could aid in fungi classification. However, the usefulness and practical benefits of this information have not been verified.

**Documentation:**

Only the project description exists in the url they provide, and the baseline code and dataset are not provided yet.

**Limitations:**

The authors addressed their limitations.

**Opportunities For Improvement:**

- Are these types of metadata truly necessary data? What about baseline experiments utilizing additional metadata or multimodal data?

- Why is it necessary to know the camera attributes?

- In the few-shot subset, if there are fewer than five samples, are the number of shots provided for each sample different?

- If so, it seems the number of shots in FungiTastic–FS is different. I suggest reporting them separately and conducting a deeper analysis of this.

- Why are the values of c_u and c_p different, and what are the criteria for this?

- Does the closed-set result include a protocol of temporal image classification?

**Relation To Prior Work:**

In Table 1 and the introductory section, they discuss the previous benchmarks but not the methodology.

**Summary And Contributions:**

This paper newly introduces FungiTastic dataset for pungi image categorization. The dataset includes photographs, textual metadata, segmentation masks, satellite images, and meteorological observations. Moreover, they collected more kinds of classes and samples than previous DF20 dataset. In addition, various tasks, including closed-set, open-set recognition, and few-shot recognition, that can be performed using this dataset are introduced, and quantitative evaluations are conducted with image recognition baselines.

---

> ### Author Rebuttal · Authors · 2024-08-16
>
> We appreciate your time and effort in reviewing the manuscript and the valuable feedback you provided. Please find our response related to your comments below. Thank you also for raising the typos; it's fixed.
>
> > **C1: Only the project description exists in the url they provide, and the baseline code and dataset are not provided yet.**
>
> We apologize for the confusion regarding the provided URLs. The working [Kaggle dataset link](https://www.kaggle.com/datasets/c2a2f11e32d1dbe110965def355907965a28b05a83244285ce7b515d2f785b0f) was provided in the supplementary material. We will ensure that both links are available in the revised version of the paper. [Correct GitHub link.](https://github.com/bohemianvra/FungiTastic/) (note that there are multiple feature branches almost ready to be merged). The final version will be provided after the review process; preliminary versions of the code are already available, for example, in feature branches.
>
> > **C2: Are these types of metadata truly necessary data?**
>
> Yes. The importance of metadata was made beneficial in other work, e.g., [Metaformer](https://arxiv.org/abs/2203.02751), [Aodha et al.](https://openaccess.thecvf.com/content_ICCV_2019/html/Aodha_Presence-Only_Geographical_Priors_for_Fine-Grained_Image_Classification_ICCV_2019_paper.html), [Picek et al.](https://openaccess.thecvf.com/content/WACV2022/html/Picek_Danish_Fungi_2020_-_Not_Just_Another_Image_Recognition_Dataset_WACV_2022_paper.html), etc. The FungiTastic dataset encourages and enables research on such dependencies.
> Some references about the utility of metadata have already been provided in Table 3, and we will further clarify this in the revised paper.
>
> > **C3: What about baseline experiments utilizing additional metadata or multimodal data?**
>
> This is a valid and good point. We will add those experiments to the paper or supplementary.
> Since baseline methods for the utilization of the metadata do not require extensive training, we have already provided a table in the General Response. The Multimodal baseline will be added to the revised version of the paper.
>
> > **C4: Why is it necessary to know the camera attributes?**
>
> For instance, the white-balancing setting influences the appropriate handling of color. Each camera model has different digital signal processing routines and a new camera not represented in the training data may have worse performance due to the “domain shift.”
>
> > **C5: In the few-shot subset, if there are fewer than five samples, are the number of shots provided for each sample different? If so, it seems the number of shots in FungiTastic–FS is different. I suggest reporting them separately and conducting a deeper analysis of this.**
>
> Very good suggestion. We will report it separately in the revised paper we already provide it in the General Response.
>
> > **C6: Why are the values of c_u and c_p different, and what are the criteria for this?**
>
> The definition of those values is related to the “assumption” that eating a poisonous mushroom is higher (c_p = 100) than not reporting a rare species (c_u = 10).
>
> > **C7: Does the closed-set result include a protocol of temporal image classification?**
>
> We are unsure if we understand the question. If you ask about the results provided in Table4, we did not employ any special temporal image classification methods.
> Is that what you mean? If not, can you please be more specific?

---

> > ### Comment · Reviewer_WN3u · 2024-08-20
> > **Questions for rebuttal**
> >
> > Thanks for reporting the results for the few-shot and metadata in the general response.
> >
> > - **Camera attributes for domain shift**
> > - If the domain shift becomes an issue, the camera type will be fixed in the training set. For example, the type of camera in the driving dataset (e.g., KITTI) was fixed due to the problem of mounting the camera into the vehicle. However, as I understand, photographs weren't collected with a limited camera. I point out this issue because simply providing additional metadata that can be collected easily does not mean that "multimodal" datasets are useful. Multimodal dataset papers should be able to experiment, verify, and convince how helpful these metadata are in problem-solving.
> >
> >
> > - **Temporal image classification**
> > - Where is the result of temporal image classification referred to in Section 3.2?

---

> > > ### Author Response · Authors · 2024-08-20
> > >
> > > > **Camera attributes for domain shift: If the domain shift becomes an issue, the camera type will be fixed in the training set. For example, the type of camera in the driving dataset (e.g., KITTI) was fixed due to the problem of mounting the camera into the vehicle. However, as I understand, photographs weren't collected with a limited camera. I point out this issue because simply providing additional metadata that can be collected easily does not mean that "multimodal" datasets are useful. Multimodal dataset papers should be able to experiment, verify, and convince how helpful these metadata are in problem-solving.**
> > >
> > > We agree that we have not provided a baseline experiment that would exploit camera data, e.g., showing that improved color constancy (white balance) leads to superior fine-grained recognition performance. By publishing the data related to the acquisition device, we expect to trigger such research, which was difficult to be carried out when no large fine-grained dataset with such info was available. There is biological evidence that color influences phenotype and that it's accurate rendering in a normalized color space will help. See  [Reinert](https://esajournals.onlinelibrary.wiley.com/doi/abs/10.2307/1939146), [Goldenberg et. al](https://onlinelibrary.wiley.com/doi/full/10.1002/ece3.11627), [Oechler et  l.](https://www.frontiersin.org/journals/ecology-and-evolution/articles/10.3389/fevo.2022.829981/full), and [Krah](https://www.frontiersin.org/journals/ecology-and-evolution/articles/10.3389/fevo.2023.1326710/full).
> > >
> > > > **Temporal image classification: Where is the result of temporal image classification referred to in Section 3.2?**
> > >
> > > We discussed the term “temporal image classification” – which led to misunderstanding. We, therefore, decided to change the terminology to “chronological classification”. (classification is called chronological if all training data are older than all test data, just like in a deployed ML system in the real world (which implies that the training and test data are not drawn from the same distribution). Since all test data are time-stamped, and the test images are ordered, the dataset can be used for continual learning experiments (i.e., when the inference on an object depends not only on the training set via the learned parameters and the observation on the object itself but potentially also on all older test set observations).
> > > Even though we do not have a specific baseline, the standard closed-set experiment, Table 4 in the submitted paper, also serves as a baseline for chronological classification. If desired, baseline based on  [Sipka et al.](https://openaccess.thecvf.com/content/WACV2022/html/Sipka_The_Hitchhikers_Guide_to_Prior-Shift_Adaptation_WACV_2022_paper.html) or [Sulc et al.](https://openaccess.thecvf.com/content_ICCVW_2019/html/TASK-CV/Sulc_Improving_CNN_Classifiers_by_Estimating_Test-Time_Priors_ICCVW_2019_paper.html) can be added to camera ready.

---

> > > > ### Comment · Reviewer_WN3u · 2024-08-21
> > > > **Thanks for response**
> > > >
> > > > The authors' answers have solved my concerns to some extent, but some baseline experiments haven't been added yet (especially for multimodal, temporal image classification), so I corrected the score to 4->5.

---

### Author Rebuttal · Authors · 2024-08-16

### General Response
We thank the reviewers for their time and effort spent reviewing the manuscript and the valuable feedback provided. We will improve the work in the following ways (for details, see the responses to individual reviewers):

**URL and Repository Links**: We will update the paper to ensure that all URLs, including the Kaggle dataset and GitHub repository, are correctly linked in the main text, abstract, and datasheet. This will prevent any future confusion and make accessing the dataset and code straightforward for all readers.

**Baseline Experiments**: We understand the need for additional baseline experiments that leverage the multimodality of provided data and metadata, and we will include such experiments in the revised paper. Below, we include two tables with such results.

**Dataset Statistics and Table Clarifications**: We will provide detailed statistics on the distribution of attributes across the dataset, specifying their availability for different subsets of data. In the paper we already stated that 99% of the observations are accompanied by all of the metadata – except for the masks, which are available only for the FungiTastic–M subset.

**Contributions and Dataset Novelty**: We will restructure the contributions statement to better distinguish between the novel aspects of our dataset and the benchmarking framework. This includes merging and clarifying contributions, as well as emphasizing the novelty in data enhancement and integration that goes beyond a simple compilation of existing data (details in response to R RwjS).


**Considering the ethical reviews**, we will update and improve the licensing. The photographs have the original license (CC-BY-NC 4.0) under which they are published in the Atlas of Danish Fungi and GBIF. The code will have the MIT license (we are in the process of changing this on Git, Kaggle, etc.). For other data (e.g., satellite images and bioclimatic time series) that do not have a specified license, we will also make it available and state that the citation of a source is required. In the paper, we provide a link and cite the source according to the request from the authors and guidelines on their website.
The second point related to the privacy concerns. Since there is no need to track the observations on the user level, no personal information is included. Therefore, the data are fully anatomized. The only minor concern could be the presence of hands in the photographs, which is fully aligned with the user's consent given in the Atlas of Danish Fungi app. To the best of our knowledge, potential negative societal impacts or privacy concerns should not apply to the proposed dataset.




### Additional Experiments:

**FewShot**: As proposed by Reviewer WN3u, we provide additional evaluation related to the number of available data for few-shot classification. Note that the few-shot dataset was created from classes with less than 5 observations, where each observation consists of 1 or more images (there are 1.82 images per observation on average).

Accuracy (%):
|model|classifier|1-shot|2-shot|3-shot|4-shot|
|:----|:---------|-------------:|-------------:|-------------:|-------------:|
|bioclip|centroid|12.69|23.48|32.86|40.26|
|bioclip|nn|12.84|21.33|30.02|29.39|
|clip|centroid|3.63|7.44|12.53|12.46|
|clip|nn|4.23|5.28|9.93|11.50|
|dinov2|centroid|12.24|17.81|23.40|31.63|
|dinov2|nn|11.93|18.00|28.61|21.41|


**Metadata**: As proposed by Reviewer 3VQA and Reviewer RwjS, we include additional experiment related to metadata exploitation. We utilize a simple yet effective approach based on our previous work (see [Picek et al.](https://openaccess.thecvf.com/content/WACV2022/papers/Picek_Danish_Fungi_2020_-_Not_Just_Another_Image_Recognition_Dataset_WACV_2022_paper.pdf)). We measured the performance improvement
with all metadata types and their combinations. Overall, habitat was most efficient in improving the performance.
With the combination of Habitat, Substrate and Month, we improved the EfficientNet-B3 model’s performance on FungiTastic-Mini by 3.62%, 3.42% and 7.46% in Top1, Top3 and F1, respectively.

|Habitat|Month|Substrate|MetaSubstrate|Top1[%]|F1[%]|Top3[%]|
|-------|-----|---------|-------------|-------|-----|-------|
|𐄂|𐄂|𐄂|𐄂|_69.45_|_43.53_|_85.39_|
|✔|𐄂|𐄂|𐄂|+2.27|+3.95|+2.25|
|𐄂|✔|𐄂|𐄂|+0.87|+1.06|+0.51|
|𐄂|𐄂|✔|𐄂|+1.15|+2.33|+0.84|
|𐄂|𐄂|𐄂|✔|+0.88|+1.51|+0.57|
|✔|✔|𐄂|𐄂|+3.12|+5.99|+2.73|
|✔|𐄂|✔|𐄂|+2.99|+5.85|+2.87|
|✔|𐄂|𐄂|✔|+2.69|+5.08|+2.58|
|𐄂|✔|✔|𐄂|+1.93|+3.99|+1.48|
|𐄂|✔|𐄂|✔|+1.57|+3.22|+1.13|
|𐄂|𐄂|✔|✔|+1.20|+2.68|+0.76|
|✔|✔|✔|𐄂|+3.62|+7.46|+3.42|
|✔|✔|𐄂|✔|+3.28|+6.80|+3.11|
|✔|𐄂|✔|✔|+2.99|+5.47|+2.92|
|𐄂|✔|✔|✔|+2.04|+4.47|+1.54|
|✔|✔|✔|✔|+3.48|+7.27|+3.37|


**Segmentation**: We provide a segmentation baseline for FungiTastic-Mini based on a combination of GroundingDINO (detection with the prompt 'mushroom') and the Segment Anything Model (segmentation prompted by detected bounding boxes). The mean of per-image intersection-over-union is 81 %.

An illustration image and per-image IoU distribution can be found here: [Github](https://github.com/BohemianVRA/FungiTastic/blob/main/assets/img/rebuttal_masks.png)

**Multimodal**: We provide a performance evaluation over a set of models submitted to our annual competition, FungiCLEF.
Evaluated on the validation set, the multimodal methods based on Metaformer outperformed the single-modality models by more than 3%.

|Model|Acc. (%)|Link|
|-----|-------|----|
|VOLO|54.34|[GitHub](https://github.com/xiaoxsparraw/CLEF2023)|
|Swin Large|55.2|[GitHub](https://github.com/wolfstefan/fungi2023)|
|MetaFormer-2|58.4|[GitHub](https://github.com/RenHuan1999/FungiCLEF2023-UstcAIGroup)|

Besides, we trained a single ResNet-18 on photos and a Multimodal Models Ensemble (MMe) with three encoders: ResNet-18 for photos, Swin-v2-t for Sentinel-2 data, and MLP for contextual info, plus one transformer decoder. The MME improved Acc. by 5.4% and F1 by 4.9%.

---

### Author Response · Authors · 2024-08-29
**Rebuttal update**

Dear reviewers and AE,

since there is no notification after rebuttal edits, we just wanted to let you know we have updated the initial rebuttal with multiple baseline results, as promised in the first version.

---

### Decision · Program_Chairs · 2024-09-26

**Decision:**

Reject

**Comment:**

This paper introduces a new classification dataset named FungiTastic. FungiTastic includes images collected over twenty years for fungal records. This dataset has more reliable ground truth based on DNA sequence, while most datasets use visual similarity determined by non-expert annotators. The task includes close-set classification, open-set classification, multi-modal classification, few-shot learning, and domain shift.

This paper has mixed opinions, two negative comments, and two positive comments from the reviewers. The main concerns include (1) the usefulness of metadata (e,g, camera attributes), (2) the scenario of each task is not well-aligned with the existing tasks (e.g., "temporal classification", "domain shift", "multi-modal classification"), (3) the absence of methods or algorithms to solve the proposed task, and (4) other minor concerns (e.g., licensing, Kaggle, github codes, ...). Since there was a significant imbalance between the frequency of discussion between the authors and the reviewers (the negative side reviewers engaged much more), I carefully read all the comments, reviews, rebuttal documents, and the original submission.

In my opinion, I agree with Reviewer WN3u and RwjS; this paper needs revision to clarify their contribution in terms of the use of terminology and the evaluation protocol. I also think that the temporal division suggested in this dataset is not well explored. Even though this paper redefines the terminology "temporal classification" to "chronological classification", it will be necessary to show that the different chronology (or timestamp) can affect the classification results. For example, the authors clarified that all test samples are time-stamped and ordered. Therefore, showing the accuracy change across different timestamps would not be very difficult. This will be very critical comment, because the contribution since the previous work (DF20) looks unclear to me. DF20 already uses the DNA-seq-based labels, all metadata proposed in this paper. For example, all the observations for metadata is already proven by the DF20 paper before.

Finally, I think the authors' comment regarding the ethical reviews, contributions, dataset novelty and URLs. However, some of them (e.g., re-write contributions and dataset novelty, adding more experiments) will require a heavy revision that might need a new review round.

Overall, I think this paper needs additional revision process to clarify their contribution compared to the previous works (many temporal classification -- or "chronological classification", metadata-plenty classification datasets, multi-modal classification, domain shift, and DF20).